# Urban Scene Vectorized Modeling Based on Contour Deformation

**Lingjie Zhu** [1,2]**, Shuhan Shen** [1,2,*] 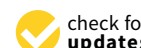**, Xiang Gao** [3] **and Zhanyi Hu** [1,2]

[1]  National Laboratory of Pattern Recognition, Institute of Automation, Chinese Academy of Sciences, Beijing 100190, China;  lingjie.zhu@nlpr.ia.ac.cn (L.Z.); huzy@nlpr.ia.ac.cn (Z.H.)
[2]  School of Artificial Intelligence, University of Chinese Academy of Sciences, Beijing 100049, China
[3]  College of Engineering, Ocean University of China, Qingdao 266100, China; xgao@ouc.edu.cn
*  Correspondence: shshen@nlpr.ia.ac.cn

**Abstract:** Modeling urban scenes automatically is an important problem for both GIS and nonGIS specialists with applications like urban planning, autonomous driving, and virtual reality. In this paper, we present a novel contour deformation approach to generate regularized and vectorized 3D building models from the orthophoto and digital surface model (DSM).The proposed method has four major stages: dominant directions extraction, find target align direction, contour deformation, and model generation. To begin with, we extract dominant directions for each building contour in the orthophoto. Then every edge of the contour is assigned with one of the dominant directions via a Markov random field (MRF). Taking the assigned direction as target, we define a deformation energy with the Advanced Most-Isometric ParameterizationS (AMIPS) to align the contour to the dominant directions. Finally, the aligned contour is simplified and extruded to 3D models. Through the alignment deformation, we are able to straighten the contour while keeping the sharp turning corners. Our contour deformation based urban modeling approach is accurate and robust comparing with the state-of-the-arts as shown in experiments on the public dataset.

**Keywords:** urban modeling; vectorization; deformation based modeling; local injective mappings

## 1. Introduction

In recent decades, there has been significant development in remote sensing technology. From valuable satellites to low-cost unmanned aerial vehicles (UAV), from high-resolution multi-spectrum images to LiDAR, we can capture both 2D and 3D data of our environment easily. Therefore, extracting semantic, geometric, and topological information from these heterogeneous data automatically has been an important research field in both GIS and nonGIS communities. Obtaining precise and compact geometry representation of large-scale urban scene is one of the core problems in urban reconstruction [1], which is referred to as vectorized modeling by many researchers. It not only has direct application in GIS like urban planning, navigation, and real estate but also is beneficial for model storage, transmission, and rendering. Apart from that, the structured and vectorized representation of an object is also a popular demand in 3D computer vision. In this paper we aim to generate compact building models from the orthophoto and digital surface model (DSM) as shown in Figure 1.

Aerial images and LiDAR point clouds are two common data types for mapping the outdoor environment. LiDAR can measure distance with great precision but the irregular form of point cloud brings an extra burden to further processing. With the increasing amount of literature on deep learning, semantic information can be easily extracted from images. Meanwhile, structure-from-motion (SfM)

and multi-view-stereo (MVS) enable us to reconstruct the scene from images with ease. Therefore, we choose the orthophoto and DSM generated from aerial images as input.

Semantic segmentation [2,3] is the fundamental part in our system. With the help of deep learning, geographic information practitioners are able to predict semantic of aerial images effortlessly. DeepLab [4,5] is one of the most acknowledged neural network structures with dilated convolution, spatial pyramid pooling, and encoder-decoder. Liu et al. [3] use a self-cascaded architecture and achieved the highest overall accuracy on the ISPRS 2D semantic labeling contest [6]. Typically the boundary of each semantic region is a chain of rasterized pixels that is too dense for a modern GIS system. The Ramer–Douglas–Peucker (RDP) algorithm [7] is widely used to simplify a polyline. It approximates a curve composed of line segments with fewer points by decimating the endpoint with minimum distance to the polyline iteratively. Poullis et al. [8] use a Gaussian mixture model and Markov random field (MRF) to classify the boundary points before applying the RDP algorithm.

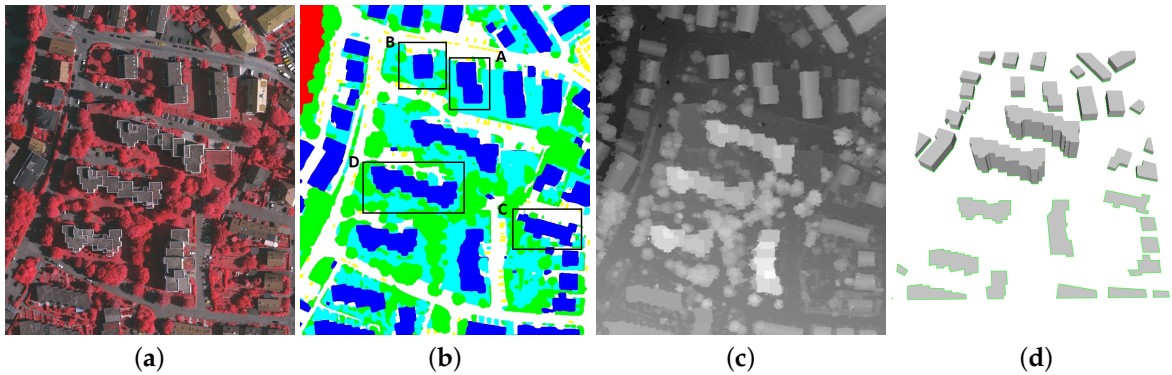

(**a**)  (**b**)  (**c**)  (**d**)

**Figure 1.** Introduction to our contour deformation based vectorized urban modeling. (**a**) is the orthophoto with RGB and infra-red channels from ISPRS [6]. (**b**) is the segmentation result for (**a**) using [3], notice the blue region represents the building. (**c**) is the DSM of the area. (**d**) shows the output level of details (LOD0) (lower half) and LOD1 (upper half) models from (**c**) in 3D.

3D geometry is another important aspect for our system. Modern SfM and MVS pipelines [9–16]. can generate accurate models from images. However, the dense triangle surface meshes or point clouds are not suitable for modern GIS system. Turning these dense outputs into the compact form, also known as vectorization has been attracting increasing attention [17–23]. Generally, they can be classified into two categorizes: (1) Bounding volume slicing and selection [17,18,24], which slice the bounding volume with planar primitives and select the polytopes inside the building or the faces on the building. (2) Contour regularization and extrusion [19,21,25,26], which usually regularize the contours of a building first with the assumption of 2.5D scene, then simplify the contours and extrude them into 3D space. Level of details (LODs) defined by CityGML [27] is widely recognized for how urban environment should be described, represented, and exchanged in the digital world. Both [17,21] focus on generating models adhering to the CityGML [27] standard. We follow the second contour regularization path and generate LOD0 and LOD1 models as shown in Figure 1d.

In this paper, we propose a novel contour deformation based urban vectorized modeling method as shown in Figure 1. We take segmented orthophoto as input and extract the contours of each building first. Then the contour normals are smoothed with a bilateral filtering, and dominant directions are detected from them with the RANSAC algorithm. After that, each edge is assigned with one of the dominant directions through an MRF. By defining a deformation energy on the triangulation of the contour polygon, we could align the boundary edges to the dominant directions. Finally, the contour can be vectorized into the polygon LOD0 model and extruded to the LOD1 model from DSM. Our main contributions include:

- An effective bilateral smoothing and RANSAC based dominant direction detection method.

- An efficient deformation energy optimization defined on the contour triangulation to align the boundary to the target directions.
- A novel deformation based building modeling method, which enables us to generate compact LOD0 and LOD1 models from orthophoto and DSM.

## 2. Proposed Method

### 2.1. Overview

The proposed modeling method takes the orthophoto and DSM as input and outputs vectorized LOD0 and LOD1 models. As shown in Figure 2, our algorithm has four major stages:

- Firstly, dominant directions of the building contour are detected through the RANSAC on the bilaterally smoothed normals, Figure 2b.
- Then each edge of the contour is assigned with one of the dominant directions as the alignment target through an MRF formulation, Figure 2c.
- With the target direction and the deformation energy defined on the contour triangle mesh, we align the boundary edges to the target direction, Figure 2d.
- Finally, compact LOD0 and LOD1 models are generated by connecting the corner vertexes and extruding them to their averaged heights in DSM , Figure 2e,f.

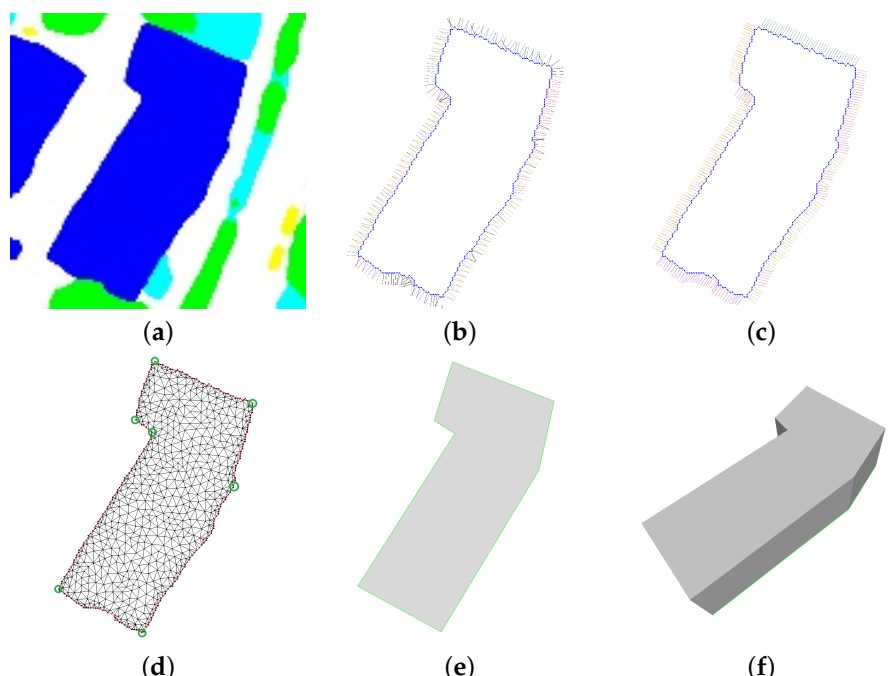

**Figure 2.** Overview of the proposed method of contour deformation based urban modeling. (**a**) is the input segmentation of a building from Figure 1. (**b**) detects the dominant directions from the bilaterally smoothed normals using RANSAC, five dominant directions are detected (color coded). (**c**) each edge is assigned with a target direction to align to with a Markov random field (MRF) formulation. (**d**) aligns the contour to the detected directions through our deformation optimization. (**e**,**f**) the generated vectorized building LOD0 model of polygon and LOD1 model extruded to the averaged height in DSM.

Generally, an urban scene has elements including road, building, and vegetation [17,21]. Among them, building is the most important category for urban vectorized modeling. Therefore, we only focus on the building in this paper. In the following sections, each building is reconstructed separately with its related data isolated from the orthophoto and DSM.

## 2.2. Dominant Directions Detection

Strong regularity in direction is one common property of the building. Manhattan assumption is widely used [24,26] in urban modeling, which assumes that the whole scene has three global orthogonal directions. The less restricted Atlanta assumption [28] requires that each building has its own local orthogonal directions. Here we do not pose any restriction to the dominant directions and detect them on the bilaterally smoothed contour normals with the RANSAC algorithm.

Figure 3a shows the building contour $\mathcal{C} = \{c_i\}$ extracted from the segmentation in Figure 1. Due to the rasterization of the image, the boundary normals $\{n_i\}$ of all boundary edges $\{e_i = \overrightarrow{c_i c_{i+1}}\}$ are limited to a few discretized directions [8]. To alleviate this problem, we propose a bilateral smoothing on the normals by weighing on both the angle similarity and the geodesic distance:

$$\hat{n}_i = \frac{\sum_{n_k \in \mathcal{N}_i} w(i,k) n_k}{\sum_{n_k \in \mathcal{N}_i} w(i,k)},\tag{1}$$

where $\mathcal{N}_i = \{n_k \mid dist(e_i, e_k) < thre_d\}$ is the neighboring normals and $dist(e_i, e_k)$ measures the distance between $e_i$ and $e_k$ along the contour. The composite weight $w(i,k)$ is given by:

$$w(i,k) = exp\left(-\frac{dist(e_i, e_k)}{\sigma_d^2} - \frac{\angle(n_i, n_k)}{\sigma_a^2}\right),\tag{2}$$

where $\angle(n_i, n_k)$ represents the angle between $n_i$ and $n_k$. In our experiments, the distance variation $\sigma_d$ and the angle variation $\sigma_a$ is set to 0.5 m and $10°$ respectively. Figure 3b shows the effect of our distance and angle weighted bilateral smoothed normals.

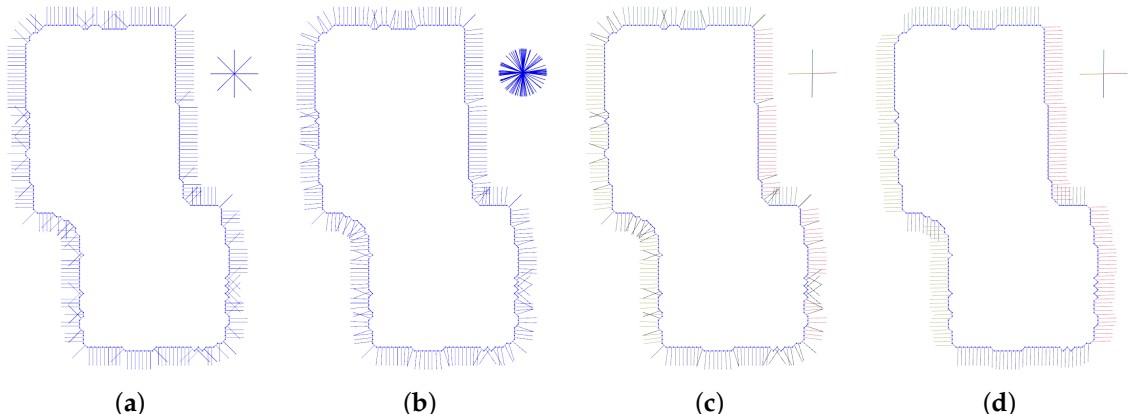

(a)　　　　　　　　　　(b)　　　　　　　　　　(c)　　　　　　　　　　(d)

**Figure 3.** Dominant directions detection and alignment direction. (**a**) initial normals orthogonal to the edges have limited variation due to rasterization. (**b**) bilaterally smoothed ($thre_d$ = 0.5 m) normals capture the dominant directions in Section 2.2. (**c**) detected four dominant directions (outliers in black) using RANSAC in Section 2.2. (**d**) find the target alignment direction for each edge with the MRF formulation in Section 2.3.

With the smoothed contour normals, the RANSAC algorithm is used to detect the dominant directions $\mathcal{D} = \{d_i\}$ with the inlier set defined as: $\mathcal{I}_i = \{\hat{n}_k \mid \angle(\hat{n}_k, d_i) < thre_a\}$. Specifically, in the $i$th iteration and with the remaining normals $\mathcal{R}_i = \{\hat{n}_k\} \setminus \bigcup_{k=1}^{i-1} \mathcal{I}_k$, we keep generating candidate directions $\{d_i^c\}$ and collecting their corresponding inlier sets $\{\mathcal{I}_i^c\}$ until the missing probability [29] $p_m$ is below a threshold $thre_p$:

$$p_m = \left(\frac{max(\{|\mathcal{I}_i^c|\})}{|\mathcal{R}_i|}\right)^{|\{d_i^c\}|}.\tag{3}$$

The best candidate $\mathcal{I}_i = argmax(\{|\mathcal{I}_i^c|\})$ is selected to compute the $i$th dominant direction:

$$d_i = \frac{\sum_{\hat{n}_k \in \mathcal{I}_i} \hat{n}_k}{\| \sum_{\hat{n}_k \in \mathcal{I}_i} \hat{n}_k \|}. \tag{4}$$

In our experiment, the iteration stops when $|\mathcal{I}_i|$ is less than $|\{n_i\}|/16$. Figure 3c shows the detected directions from the smoothed contour normals.

*2.3. Align Direction*

With the detected dominant directions $\mathcal{D}$, we adopt the similar MRF formulation in [8] to assign each boundary edge $e_i$ a target direction $d_i$, which will be later used to drive the deformation in the following Section 2.4.

For each building contour $\mathcal{C}$, an undirected dual graph $\mathcal{G} = \langle \mathcal{V}, \mathcal{E} \rangle$ is constructed on it, meaning each edge $e_i$ is treated as a vertex in $\mathcal{V}$ and each vertex $c_i$ is considered as an edge in $\mathcal{E}$. Accordingly, our label set is $\mathcal{D} = \{d_i\}$ and a labeling configuration is $f : \mathcal{V} \to \mathcal{D}$. The data term measures the difference between the observed data $\hat{n}_i$ and the mapped label $f(e_i)$:

$$E_{data}(f) = \sum_{e_i \in \mathcal{V}} \| \hat{n}_i - f(e_i) \|. \tag{5}$$

The smoothness term favors the connecting edges $e_i, e_j \in \mathcal{V}$ with similar orientations $f(e_i), f(e_j) \in \mathcal{D}$ and penalizes otherwise:

$$E_{smooth}(f) = \sum_{\langle e_i, e_j \rangle \in \mathcal{E}} \left( \sqrt{2} - e^{-\frac{\| \hat{n}_i - \hat{n}_j \|}{2\sigma^2}} \right) \cdot \| f(e_i) - f(e_j) \|, \tag{6}$$

where $\delta$ controls the smoothness uncertainty. Intuitively, if two neighboring edge normals $\hat{n}_i$ and $\hat{n}_j$ are close, there is a higher probability that $f(e_i)$ and $f(e_j)$ are similar. The last label term favors a lower number of labels in a labeling configuration $f$:

$$E_{label}(f) = \sum_{d_i \in \mathcal{D}} (1 - h_i)^2 \cdot \zeta_i(f), \tag{7}$$

where $h_i = |\mathcal{I}_i| / \sum_{i=1}^{|\mathcal{D}|} |\mathcal{I}_i|$ measures the relative portion of a dominant direction on the contour and $\zeta_i(f)$ indicates whether label $d_i$ exist in $f$:

$$\zeta_i(f) = \begin{cases} 1, & \exists e_k : f(e_k) = d_i, \\ 0, & \text{otherwise.} \end{cases} \tag{8}$$

The label term penalizes heavily when there exists a label that corresponds to a direction of little portion $h_i$, thus tends to keep the few most significant directions in $\mathcal{D}$. Then the overall energy function for the graph-cut is given by:

$$E(f) = E_{data}(f) + \kappa_1 E_{smooth}(f) + \kappa_2 E_{label}(f), \tag{9}$$

where $\kappa_1$ and $\kappa_2$ are the balance coefficients of the smoothness term and the label term respectively. They are set to 1 and 10 in the following section. As proven in [8], the formulation is regular and can be minimized by the $\alpha$-expansion algorithm [30,31]. Figure 3d shows how each boundary edge is assigned with a target dominant direction.

*2.4. Deformation Formulation*

This section introduces our dominant direction driven deformation formulation, which regularizes the contour and greatly helps the following modeling step. For each contour $\mathcal{C}$, we use the constrained

Delaunay triangulation [32] to triangulate the bounded area as shown in Figure 4. The generated mesh $\mathcal{M} = \{\mathbf{t}_1, \cdots, \mathbf{t}_l\}$ contains vertexes $\{\mathbf{v}_1, \cdots, \mathbf{v}_m\}$ and boundary edges $\{e_1, \cdots, e_n\}$.

Our goal is to align the normal $\hat{n}_i$ of each $e_i$ on $\mathcal{C}$ to its assigned target direction under the configuration $f$ in Section 2.3. Therefore, we have the align energy that measures the difference between the edge normals $\hat{n}_i$ and their target dominant directions $d_i \in \mathcal{D}$:

$$E_{align} = \sum_{i=1}^{|\mathcal{C}|} (\|e_i\|\hat{n}_i \cdot f(e_i))^2. \tag{10}$$

By changing the positions of $\{\mathbf{v}_1, \cdots, \mathbf{v}_m\}$, we can deform $\mathcal{M}$ to align the contour $\mathcal{C}$ to major directions $\mathcal{D}$ and minimize $E_{align}$. The deformation is a piece wise linear function $g : \mathbb{R}^2 \to \mathbb{R}^2$ defined on $\mathcal{M}$, which maps every original triangle $\mathbf{t} = \triangle \mathbf{v}_p \mathbf{v}_q \mathbf{v}_r$ to the output triangle $\mathbf{t}' = \triangle \mathbf{v}'_p \mathbf{v}'_q \mathbf{v}'_r$ as illustrated in Figure 4. We adopt the Advanced Most-Isometric ParameterizationS (AMIPS) [33] to achieve as low distortion as possible. Then the mapping is an affine transformation defined on each $\mathbf{t}$: $g_{\mathbf{t}}(x) = J_{\mathbf{t}}x + b_{\mathbf{t}}$, where $J_{\mathbf{t}}$ is a $2 \times 2$ affine matrix and also the Jacobian [33] of $g_{\mathbf{t}}$:

$$J_{\mathbf{t}} = \left[ \mathbf{v}'_p - \mathbf{v}'_q, \mathbf{v}'_p - \mathbf{v}'_r \right] \cdot \left[ \mathbf{v}_p - \mathbf{v}_q, \mathbf{v}_p - \mathbf{v}_r \right]^{-1}. \tag{11}$$

With the singular value of $J_{\mathbf{t}}$ denoted as $\sigma_1$ and $\sigma_2$, the AMIPS [33] conformation and area distortion energy defined on $\mathbf{t}$ is:

$$\delta_{conf,\mathbf{t}} = \frac{\sigma_1}{\sigma_2} + \frac{\sigma_2}{\sigma_1} = \frac{\text{trace}(J_{\mathbf{t}}J_{\mathbf{t}}^T)}{\det J_{\mathbf{t}}}, \tag{12}$$

$$\delta_{area,\mathbf{t}} = (\det J_{\mathbf{t}} + (\det J_{\mathbf{t}})^{-1}). \tag{13}$$

The rigid energy measuring how rigid the mapping from the original triangle $\mathbf{t}$ to the deformed triangle $\mathbf{t}'$ is defined as:

$$E_{rigid} = \sum_{t \in \mathcal{M}} exp(\alpha \delta_{conf,\mathbf{t}} + (1 - \alpha)\delta_{area,\mathbf{t}}). \tag{14}$$

Then the overall deformation energy balanced by $\lambda$ is

$$E_{deform} = E_{rigid} + \lambda E_{align}. \tag{15}$$

To minimize $E_{deform}$, we use the MATLAB optimization toolbox and the closed form gradient is in the Appendix A. The conformation and area distortion balance $\alpha$ is set to 0.5 and the alignment and deformation balance $\lambda$ is set to $1e5$. Figure 5 shows the process of energy minimization.

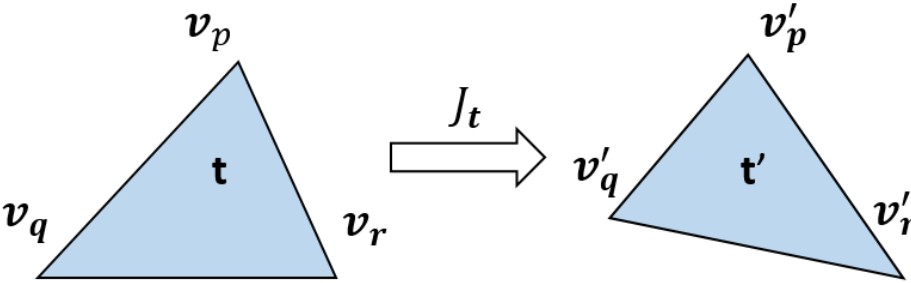

**Figure 4.** Mapping of a triangle $\mathbf{t}$ to the deformed triangle $\mathbf{t}'$ through an affine transformation $J_{\mathbf{t}}$.

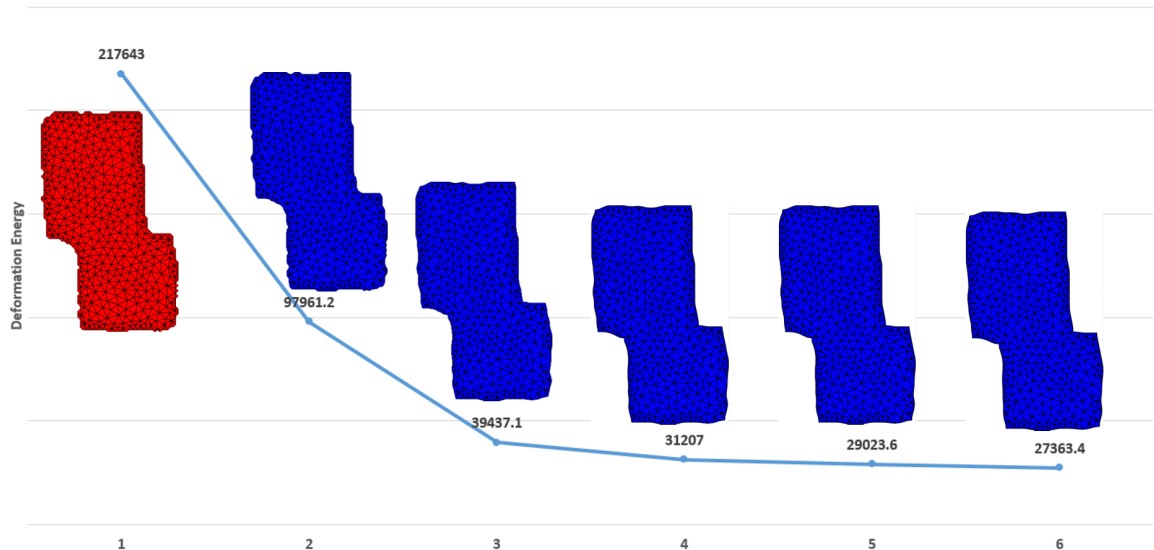

**Figure 5.** Optimization of the energy $E_{deform}$. While the energy drops at each iteration, the contour is gradually aligned to its assigned target direction in Section 2.3.

### 2.5. Model Generation

After the deformation optimization in Section 2.4, the building contour $\mathcal{C}$ becomes $\mathcal{C}'$ and is aligned to the dominant directions. LOD0 and LOD1 models adhering to the CityGML [27] standard can be easily extracted from it. Specifically, we traverse the contour $\mathcal{C}'$ and only keep the corner vertexes $\{c_i'\}$, which has large angle between the neighboring edges $e_{i-1}'$ and $e_{i+1}'$. Connecting the corner vertexes gives us the LOD0 model of a simplified polygon. LOD1 model is given by extruding the LOD0 contour to the averaged height of the building [17,21]. Figure 6 shows the process of model generation.

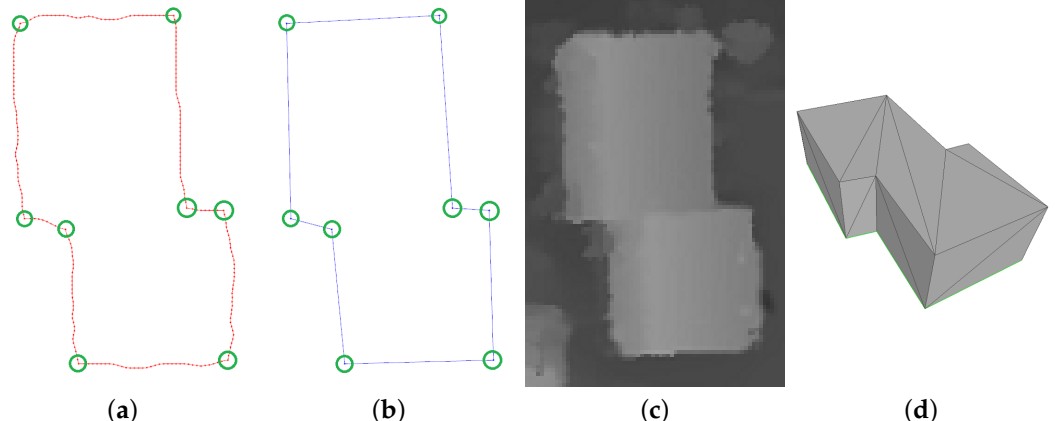

| (a) | (b) | (c) | (d) |

**Figure 6.** Generating LOD models from the deformed contour in Figure 5. (**a**) is the deformed contour with corner vertexes in green circles. (**b**) is the LOD0 model of simplified polygon reducing the vertexes from 499 to 8. (**d**) is the LOD1 model by extruding the contour to the averaged height in (**c**).

### 3. Results and Discussion

The proposed method is implemented in C++. The max-flow library [30,31] is used to solve the MRF in Section 2.3 and the MATLAB optimization toolbox is used to solve the energy minimization in Section 2.4. Qualitative as well as quantitative assessments are conducted on the public dataset from ISPRS [6] in this section. Experiments show that our contour deformation optimization framework can generate regular and compact building models compared to the state-of-the-arts.

### 3.1. Effect of Alignment Deformation

To demonstrate the effect of the alignment deformation, we conduct a simple experiment that aligns the contour to the axes, as shown in Figure 7. The contour is extracted from our own real world orthophoto of a high rising building, which is sampled from the dense textured mesh reconstructed from aerial images by Pix4D [34]. Due to the high level of noise and inaccuracy derived from the mesh reconstruction and orthophoto segmentation, the extracted building contour is uneven and zigzags as illustrated in Figure 7. To make matters worse, the corners of small protrusion are rounded (green rectangle in Figure 7), which is essential for vectorized modeling.

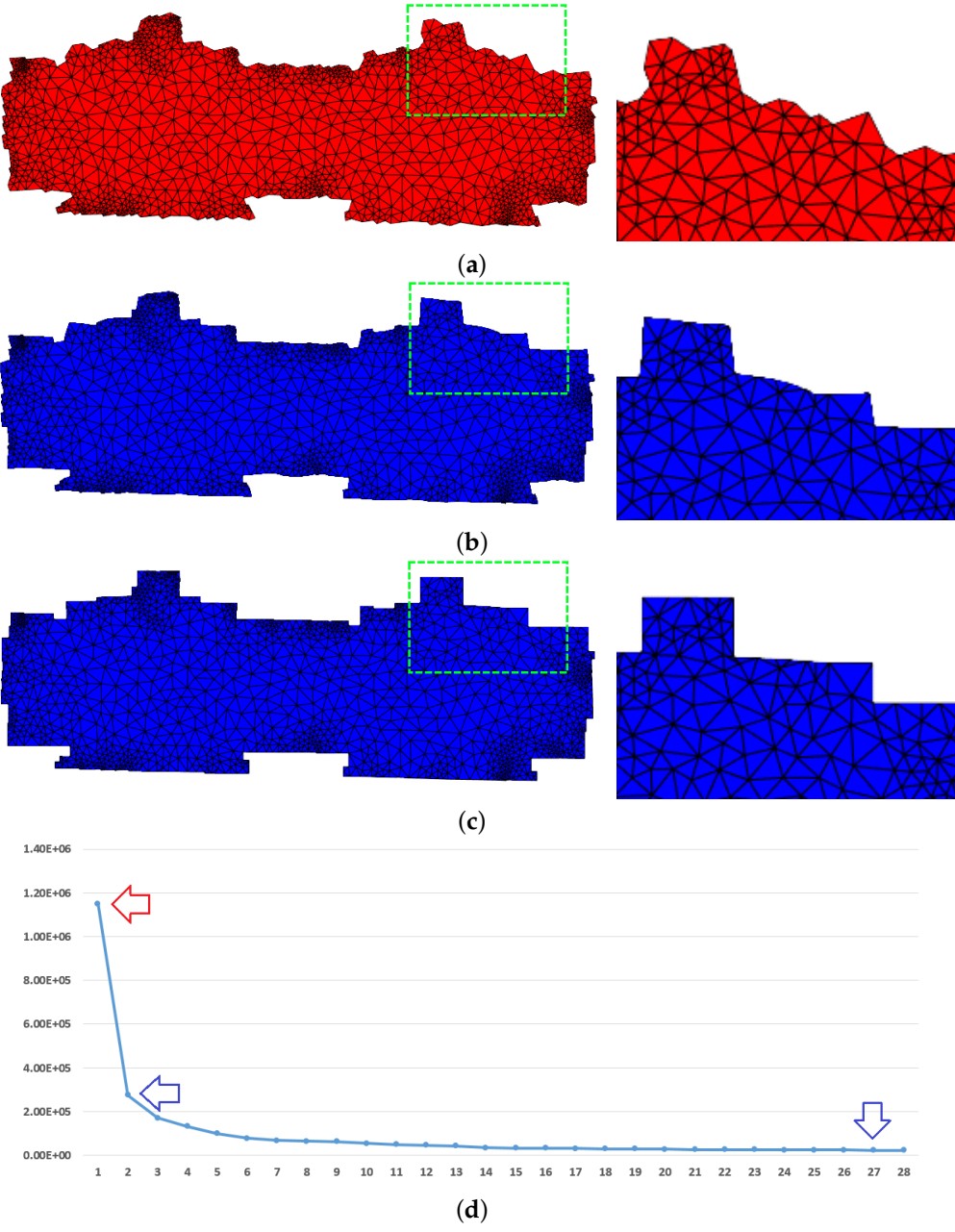

**Figure 7.** Deformation energy minimization aligning to the nearest axes on a large contour. The right side of the first three rows is the close-ups of the area in the green rectangles on left. (**a**) The input jaggy contour triangle mesh has 1424 vertexes, 2508 faces and 338 boundary edges. The initial energy is $1,150,510$ (**b**) The contour mesh at the 2nd iteration with energy of $170,687$. (**c**) The contour mesh at the 27th iteration with energy of $23,589$. (**d**) The energy minimization curve from the first iteration to the last one. It drops rapidly in the first few iterations and stops eventually.

For each border edge normal, we simply assign the nearest axis as its target direction. As shown in Figure 7 the jaggy input contour is aligned to the axes gradually as the deformation energy drops. The closeup on the right of the area in the green rectangle on the left shows detail of the optimization process. The energy drops dramatically in the first few iterations as shown in Figure 5d. Since we set the alignment and rigidness balance $\lambda$ to $1 \times 10^5$, we can also observe that the alignment is reached quickly at first, then the rigidness in the following iterations.

### 3.2. Quality Comparison

There are three major aspects in evaluating the modeling quality: contour accuracy, contour complexity and regularity. Both general urban modeling algorithms [8,21] and general polyline simplification algorithm [7] are evaluated and compared on the public Vaihingen data set [6]. The input orthophotos shown in Figures 1 and 8 are generated via dense image matching with Trimble INPHO 5.3 software and Trimble INPHO OrthoVista [6]. The sampling step of both the TOP and the DSM is 9 cm in Figures 1 and 8.

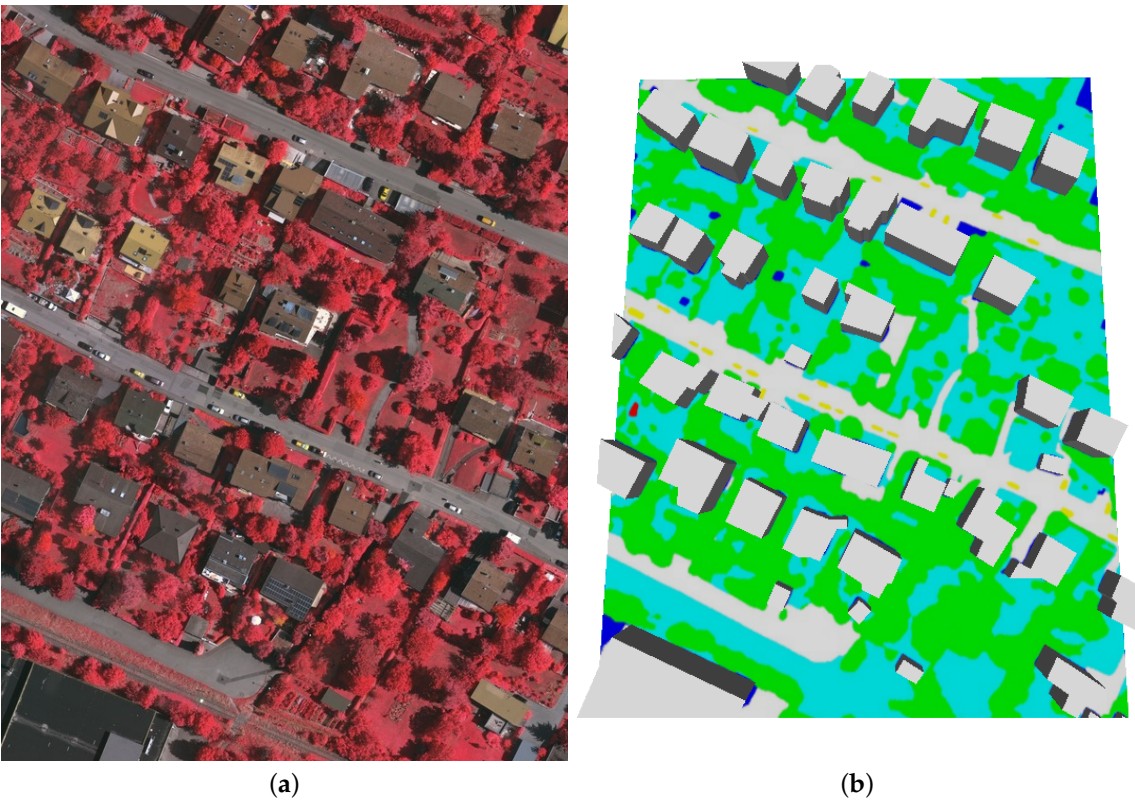

(**a**) (**b**)

**Figure 8.** Modeling result on another block. (**a**) The input orthophoto with infra-red channel of another urban area. (**b**) Our generated LOD1 model of the scene overlaid on the segmentation.

For LOD0 and LOD1 generation, the problem is usually treated as a contour extraction and simplification problem. The RDP algorithm [7] removes the vertex close to the curve iterative until the target number of vertexes is reached or the minimum distance is reached. The simplified curve is a subset of the points that are in the original polyline. This makes it easily affected by the few extreme points as shown in Table 1.

Poullis et al. [8] conduct a orientation detection and classification before the RDP algorithm [7]. This enables it to be aware of the turning points of the global orientation. Unfortunately, the simplified curve is still the subset of the input curve. Therefore, RDB [7] and Poullis et al. [8] could not capture the position of the corners well, due to rounded corners in the segmentation mask as shown in Table 1.

**Table 1.** Contour quality comparison of different methods on the building in Figure 3. In each row from left right to left are: the ground truth building contour provided by ISPRS [6], the our LOD0 model, the RDP algorithm [7] output by specifying the number of vertexes to be the same as ours, the result by Poullis et al. [8], and Zhu et al. [21].

| Ground Truth | Ours LOD0 | RDP [7] | Poullis et al. [8] | Zhu et al. [21] |
|:---:|:---:|:---:|:---:|:---:|
| IoU | **0.96** | 0.94 | 0.95 | 0.87 |
| IoU | **0.94** | 0.92 | 0.93 | 0.78 |

The last column in Table 1 demonstrates that Zhu et al. [21] could generate contour with strong regularity. However, it fails when the scene does not satisfy the Manhattan assumption. The missing part in the lower left corner of the second building in Table 1 is caused by the missing short segments.

As illustrated in Table 1, our method has the highest IoU while maintaining the sharp corners of the contours. Thanks to the deformation optimization process, we are able to recover the position of the corner vertex more accurately. Table 2 lists more results of the buildings in Figure 1, and the orthogonal neighboring edges are correctly reconstructed. In addition, Figure 8 shows the result on another block from the Vaihingen data set [6]. On the left is the input orthophoto of residential area, and on the right is the output LOD1 model overlaid on the semantic segmentation map [3].

**Table 2.** Generated model on different buildings in Figure 2. In each column from top to bottom are: building segmentation, generated vectorized polygon LOD0 model, and the extruded LOD1 model.

| Region | B | C | D |
|--------|---|---|---|
| Contour | | | |
| LOD0 | | | |
| LOD1 | | | |

## 4. Conclusions

In this paper, we try to turn the dense 2D orthophotos into compact 3D polygon models automatically for the urban scene [17,21], which is suitable for efficient representation, processing, and rendering of large-scale scenes. The generated models can be used in various fields like urban planning, navigation, emergency simulation, and risk reduction [1].

Specifically, we propose a novel deformation based contour simplification approach that generates vectorized LOD0 and LOD1 building models from the orthophotos. To begin with, building contours are extracted from the semantic labeled orthophotos and processed separately. For each building, we first extract dominant directions by applying the RANSAC algorithm on the bilaterally smoothed contour edge normals. Then each edge normal is assigned with one of the dominant directions as target alignment by formulating the task as an MRF labeling problem [8]. Finally, a deformation energy combining the edge normal alignment and AMIPS [33] rigidness is defined on the contour triangle mesh. By minimizing the deformation energy and connecting the corner vertexes, we could generate compact LOD0 and LOD1 models easily.

Compared to the classic RDP algorithm [7] and the most recent advanced contour based methods [8,21], we are able to enhance the global regularity and retain the contour topology at the same time. The proposed novel deformation approach could aggregate different constraints into one optimization framework. It shows great potential in the urban scene modeling, which is rich with regularities like orthogonality, parallelism, and collinearity. In the future, we would like to add more common regularity constraints and take the generated models to higher LODs.

**Author Contributions:** Methodology, Lingjie Zhu and Shuhan Shen; software, Lingjie Zhu; validation, Xiang Gao; investigation, Xiang Gao; resources, Shuhan Shen; data curation, Shuhan Shen; writing—original draft preparation, Lingjie Zhu; writing—review and editing, Shuhan Shen and Xiang Gao; visualization, Lingjie Zhu;

supervision, Zhanyi Hu; project administration, Zhanyi Hu; funding acquisition, Zhanyi Hu. All authors have read and agreed to the published version of the manuscript.

**Funding:** This research was funded by Natural Science Foundation of China under Grants 61991423, 61873265, 61421004.

**Acknowledgments:** We would like to acknowledge the public data set retained by ISPRS [6].

**Conflicts of Interest:** The authors declare no conflict of interest.

## Appendix A

The essential part of the deformation energy $E_{deform}$ is the MIPS energy $\delta_{conf,\mathbf{t}}$ and the area energy $\delta_{area,\mathbf{t}}$ defined on each triangle $\mathbf{t}$. According to [33], their derivatives with respect to a variable $x$ are:

$$\partial_x \delta_{conf,\mathbf{t}} = \frac{2trace(J_{\mathbf{t}}^T \cdot \partial_x J_{\mathbf{t}})}{det J_{\mathbf{t}}} - \delta_{conf,\mathbf{t}} trace(J_{\mathbf{t}}^{-1} \cdot \partial_x J_{\mathbf{t}}), \tag{A1}$$

$$\partial_x \delta_{area,\mathbf{t}} = (det J_{\mathbf{t}} - det J_{\mathbf{t}}^{-1}) trace(det J_{\mathbf{t}}^{-1} \cdot \partial_x J_{\mathbf{t}}). \tag{A2}$$

The overall derivatives can be obtained easily by the chain rule.

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
