# Peer review of "Urban Scene Vectorized Modeling Based on Contour Deformation"

_ijgi, doi:10.3390/ijgi9030162_

Round 1

Reviewer 1 Report

This is a very interesting approach to displaying buildings in a GIS environment.  I think the authors have a solid grasp on the relevant literature, sound methods and the results are useful to GIS practitioners.  I have two concerns with the article.  First, the article will require significant editing of grammar and usage.  I had trouble following your discussion at times, especially in the abstract and introduction, but the entire paper had grammatical and usage errors. 

Second, I would like to see a very brief discussion in the abstract, introduction and conclusion that addresses and explains your research problem in less specific terminology.  To phrase it another way, why would the non-GIS specialist want to read your article?  I think the idea of modeling building shapes is an extremely important concept for urban planners, urban geographers and others, I just want to see something that gets them excited about the paper.  I understand this article will speak to the specialist, but I think your message is important enough to include some general statements for the non-GIS specialist. 

Congratulations on a wonderful research paper. 

Author Response

Response to Reviewer 1 Comments

Point 1: First, the article will require significant editing of grammar and usage. I had trouble following your discussion at times, especially in the abstract and introduction, but the entire paper had grammatical and usage errors.

Response 1: Thanks for your kind reminder about our grammar and usage errors. We have revised our manuscript accordingly with the help of a native English speaker.

Point 2: Second, I would like to see a very brief discussion in the abstract, introduction and conclusion that addresses and explains your research problem in less specific terminology.  To phrase it another way, why would the non-GIS specialist want to read your article?  I think the idea of modeling building shapes is an extremely important concept for urban planners, urban geographers and others, I just want to see something that gets them excited about the paper.  I understand this article will speak to the specialist, but I think your message is important enough to include some general statements for the non-GIS specialist.

Response 2: Thanks for your constructive suggestion. In the revised version, we have added a few words in the abstract, introduction and conclusion to make the research problem clearer and more attractive to the non-specialists like urban planners and urban geographers. The abstraction starts with: modeling urban scenes automatically is an important problem for both GIS and non-GIS specialists with applications like urban planning, navigation, and simulation. In conclusion: In this paper, we try to turn the dense 2D orthophotos into compact 3D polygon models automatically for the urban scene [17,21], which are suitable for efficient representation, processing, and rendering of large-scale scenes. The generated models can be used in various fields like urban planning, navigation, emergency simulation, and risk reduction.

Reviewer 2 Report

Dear Respected Editor,

I have read this paper in a detailed manner. Authors, in this paper, have presented a novel contour deformation approach to generate regularized and vectorized 3D models from the orthophoto and DSM data. They have considered the proposed method which four major stageswith dominant directions extraction, find target align direction, contour
deformation, and model generation. They have extracted dominant directions for each building. They have introduced their ideas via figures.

This paper is we written in English, but, there are some modifications defined as follows:

1- E-mail adresses of Authors must be corrected accordingly.
2-All paper must be checked grammatically because it includes some modifications.
3-Ä°t is of 9% similarity. Thus, it is an interesting paper. Congrats to Authors.
4-Conclusion section is too short. They need to extend it by adding some novelties of the paper.
5-The authors are request to add more details regarding their original contributions in this manuscript.
6. Papers cited in references section is almost very very old. Hence, with some papers published newly, it should be revised.

Briefly, this paper is well written in English. Therefore, It may be accepted to publish after doing necessary revisions defined above.
With my best regards

Author Response

Response to Reviewer 1 Comments

Point 1: E-mail adresses of Authors must be corrected accordingly.

Response 1: Corrected in the revised version.

Point 2: All paper must be checked grammatically because it includes some modifications.

Response 2: Checked and corrected in the revised version.

Point 3: Ä°t is of 9% similarity. Thus, it is an interesting paper. Congrats to Authors.

Response 3: Thanks.

Point 4: Conclusion section is too short. They need to extend it by adding some novelties of the paper.

Response 4: The revised conclusion section is extended with an address of the novelties of the paper. In this paper, we try to turn the dense 2D orthophotos into compact 3D polygon models automatically for the urban scene [17,21], which are suitable for efficient representation, processing, and rendering of large-scale scenes. The generated models can be used in various fields like urban planning, navigation, emergency simulation, and risk reduction [1]. Specifically, we propose a novel deformation based contour simplification approach that generates vectorized LOD0 and LOD1 building models from the orthophotos. To begin with, building contours are extracted from the semantic labeled orthophotos and processed separately. For each building, we first extract dominant directions by applying the RANSAC algorithm on the bilaterally smoothed contour edge normals. Then each edge normal is assigned with one of the dominant directions as target alignment by formulating the task as a MRF labeling problem [8]. Finally, a deformation energy combining the edge normal alignment and AMIPS [33] rigidness is defined on the contour triangle mesh. By minimizing the deformation energy and connecting the corner vertexes, we could generate compact LOD0 and LOD1 models easily.

Point 5: The authors are request to add more details regarding their original contributions in this manuscript.

Response 5: In the revised introduction and conclusion sections, we state our original contributions in detail. 1) An effective bilateral smoothing and RANSAC based dominant direction detection method. 2) An efficient deformation energy optimization defined on the contour triangulation to align the boundary to the target directions. 3) A novel deformation based building modeling method,
which enables us to generate compact LOD0 and LOD1 models from orthophoto and DSM.

Point 6: Papers cited in references section is almost very very old. Hence, with some papers published newly, it should be revised.

Response 6: Thank you for pointing that out. We have switched our references to more recently published papers in the revised version.